# An Innovative Three-Stage Model for Prenatal Genetic Disorder Detection Based on Region-of-Interest in Fetal Ultrasound

**DOI:** 10.3390/bioengineering10070873

**Published:** 2023-07-23

**Authors:** Jiajie Tang, Jin Han, Yuxuan Jiang, Jiaxin Xue, Hang Zhou, Lianting Hu, Caiyuan Chen, Long Lu

**Affiliations:** 1Institute of Pediatrics, Guangzhou Women and Children’s Medical Center, Guangzhou Medical University, Guangzhou 510623, China; jyx19960130@163.com; 2School of Information Management, Wuhan University, Wuhan 430072, China; 3Graduate School, Guangzhou Medical University, Guangzhou 511495, China; 4Center for Healthcare Big Data Research, The Big Data Institute, Wuhan University, Wuhan 430072, China; 5Medical Big Data Center, Guangdong Provincial People’s Hospital, Guangzhou 510317, China; 6Guangdong Cardiovascular Institute, Guangdong Provincial People’s Hospital, Guangzhou 510317, China; 7School of Public Health, Wuhan University, Wuhan 430072, China

**Keywords:** fetal genetic disorder, ensemble learning, YOLOv5, CNN, region of interest

## Abstract

A global survey has revealed that genetic syndromes affect approximately 8% of the population, but most genetic diagnoses are typically made after birth. Facial deformities are commonly associated with chromosomal disorders. Prenatal diagnosis through ultrasound imaging is vital for identifying abnormal fetal facial features. However, this approach faces challenges such as inconsistent diagnostic criteria and limited coverage. To address this gap, we have developed FGDS, a three-stage model that utilizes fetal ultrasound images to detect genetic disorders. Our model was trained on a dataset of 2554 images. Specifically, FGDS employs object detection technology to extract key regions and integrates disease information from each region through ensemble learning. Experimental results demonstrate that FGDS accurately recognizes the anatomical structure of the fetal face, achieving an average precision of 0.988 across all classes. In the internal test set, FGDS achieves a sensitivity of 0.753 and a specificity of 0.889. Moreover, in the external test set, FGDS outperforms mainstream deep learning models with a sensitivity of 0.768 and a specificity of 0.837. This study highlights the potential of our proposed three-stage ensemble learning model for screening fetal genetic disorders. It showcases the model’s ability to enhance detection rates in clinical practice and alleviate the burden on medical professionals.

## 1. Introduction

Birth defects are the leading cause of disability and death among neonates and newborns, and they are also known as the “silent killers” of various human diseases [1]. The etiology of birth defects is complex and encompasses a wide range of factors, which are primarily divided into genetic and environmental causes. Genetic factors directly or indirectly account for over 80% of the etiology of birth defects [2]. Chromosomal disorders represent the most common category of genetic disorders and are a focal point in the prevention of birth defect-related diseases. To date, more than 20,000 types of human chromosomal abnormalities and structural variations have been identified, along with over 200 chromosomal syndromes [3]. The incidence of these disorders in newborns reaches as high as 8 in 100 [4]. Chromosomal disorders are classified as severe birth defects, leading to either death or significant disability, and currently, effective treatment methods are still lacking [5].

Chromosomal disorders result from abnormal genetic factors and commonly involve facial deformities in affected individuals [6]. Children with the same syndrome often exhibit similarities and specific facial features, such as Down syndrome (T21), Trisomy 18 syndrome (T18), Trisomy 13 syndrome (T13), and Cornelia de Lange syndrome (CdLS). Prenatal diagnosis plays a crucial role in preventing the birth of severely disabled children. Current ultrasound diagnostic techniques are indispensable for prenatal diagnosis, enabling the identification of abnormal fetal structural development and genetic information [7,8,9,10,11]. However, prenatal diagnosis based on ultrasound imaging faces challenges such as inconsistent diagnostic criteria, a shortage of medical personnel, a lack of dynamic monitoring methods, and limited coverage [12,13]. Consequently, genetic disorders have become one of the significant factors affecting the quality of the global population at birth. For the well-being of society, the establishment of automated diagnostic and monitoring systems for genetic disorders has become a crucial issue that the academic community must address, as it represents a pressing need for society as a whole.

The rapid advancement of deep learning technology has led to the widespread exploration and development of automatic facial recognition systems. Recent studies have demonstrated the potential of facial analysis technology to enhance clinicians’ diagnostic capabilities for genetic syndromes. Gurovich et al. created a genetic disease recognition system based on children’s facial photos [14]. Consequently, it has garnered significant attention in the field of clinical genetics. Based on this study, Porras et al. conducted a comprehensive analysis of facial features in children and utilized the aggregation of facial regions to identify the probability of developing genetic disorders [15]. This investigation revealed the significant relevance of specific facial areas, such as the nose, mouth, and forehead, in the screening of genetic disorders. As of the present, there is limited research on analyzing fetal facial images and developing artificial intelligence models. Yasunari et al. developed an artificial intelligence classifier to recognize fetal facial expressions related to fetal brain development, including blinking, mouth opening, blank expression, frowning, smiling, sticking out the tongue, and yawning [16]. This study revealed the feasibility of utilizing ultrasound imaging for the identification of fetal facial images.

We propose FGDS, an innovative three-stage ensemble learning model specifically designed for the accurate detection of genetic disorders in the prenatal stage. The FGDS model employs a comprehensive approach, beginning with the extraction of multiple regions of interest (ROI) from ultrasound images. Subsequently, convolutional neural networks (CNNs) are utilized to establish the intricate connection between these ROIs and genetic disorders. Finally, a decision layer fusion method based on XGBoost 1.4.2 is employed to effectively integrate the correlation between each ROI, thus achieving reliable and comprehensive screening of genetic disorders. We believe that FGDS leverages multiple fetal facial regions and can enhance model performance and ensure robustness. It could help with prenatal ultrasound diagnosis, reduce false-negative results, and compensate for the lack of medical resources.

## 2. Materials and Methods

The present study received ethical approval from the Institutional Review Board of the Guangzhou Women and Children’s Medical Center (Approval No. 473B01, 2021). Written informed consent was obtained from all participants prior to their involvement in the study. All ultrasound examinations were conducted by a team of specialists with over five years of experience before the commencement of the research. We excluded cases from the initial list if they lacked genetic results, ultrasound images, or complete clinical data. Only one ultrasound image is included for each examination. All cases have genetic test results, which are used as the gold standard for diagnosis [17]. The genetic disorders included in the study contained four main categories: T21, T18, T13, and other rare genetic disorders. Other genetic disorders without facial abnormalities were not included in this study.

### 2.1. Dataset Collection and Dataset Characteristics

In this study, a total of 2554 pregnant women aged between 23 and 38 years, who underwent prenatal diagnosis at our center, were included. Ultrasound fetal profile images were collected from these participants. The dataset consisted of 1493 images from normal pregnancies and 1061 images representing cases with genetic disorders. The gestational age of the included cases ranged from 11 to 27 weeks. Table 1 provides detailed information regarding the composition of the training dataset and testing set.

Table 1 presents an overview of the dataset employed in our research study. The dataset is divided into two parts: the retrospective dataset and the prospective dataset. Additionally, it provides information on the training dataset, retrospective test set, and prospective test set. The retrospective dataset consists of a total of 1780 images, while the prospective dataset contains 410 images. Within the retrospective dataset, 1094 images represent negative results, indicating the absence of the target condition, while 686 images depict positive results, representing the presence of the target condition. In the prospective dataset, there are 233 images of negative results and 177 images of positive results. The ultrasound equipment used in the study varied between the datasets. The retrospective dataset employed GE Volution E10 ultrasound equipment. In contrast, the prospective dataset involved multiple ultrasound machines, namely the GE Voluson E6, E8, E10, and Philips iE33. The study included four ultrasonographers who performed the examinations for the retrospective dataset. In the case of the prospective dataset, more than ten ultrasonographers were involved in the data collection process.

### 2.2. Data Preprocessing

The standardization of data collected in clinical practice is different from that of public datasets. Due to various external factors such as equipment, personnel, lighting, and collection methods, inconsistent image standardization may occur, and the amount of information contained in the image may be significantly reduced. In this study, the homomorphic filtering method was used to preprocess ultrasound images [18]; the filtering results are shown in Appendix A.

### 2.3. Determination of Regions of Interest (ROI)

The examination section used in this study is the median sagittal section, which includes anatomical structures such as NT (nuchal translucency), NB (nasal bone), nasal tip, jawbone, and cranial crest, which are used in clinical diagnoses of high-risk fetuses [19,20]. Considering that the nasal tip, nasal bone, maxilla, and mandible have positional relationships, this study included them in the same region of interest. In addition, because the brain contains complex positional information, it is also considered a region of interest. Although the diencephalon is not related to screening for genetic disorders, it is an important anatomical structure in this section. Additionally, our study attempts to extract this anatomical structure to explore the feasibility of extracting key anatomical structures. In summary, this section divides the regions of interest that need to be extracted based on clinical diagnostic experience, and the detailed information is shown in Figure 1.

### 2.4. Model Architecture

As shown in Figure 2, FGDS is composed of three interconnected parts. The first part, referred to as Network A, focuses on the extraction of ROI from the input data. Following that, the second part, Network B, concentrates on extracting the genetic disorder information within the identified ROIs. Finally, in the third part, denoted as Network C, we utilize the XGBoost algorithm to effectively integrate the disease-related information obtained from multiple regions of interest, enabling the estimation of the genetic disorder risk. By employing this three-part architecture, FGDS enhances the analysis of fetal facial features and provides a comprehensive approach to genetic disorder screening. Here is the mathematical expression for the three-stage ensemble learning model:

Feature extraction in Network A:(1)R=A(I)
where $R=\left\{r_{1},r_{2},\ldots ,r_{n}\right\}$ represents the set of *ROI* and n is the number of regions.

Risk estimation Network B:

For each region $r_{i}$ in R, calculate its risk value $s_{i}$:(2)$$s_{i}=B\left(r_{i}\right)$$
where i denotes the ith region.

Risk aggregation in Network C*:*(3)P=C(S)
where $S=\left\{s_{1},s_{2},\ldots ,s_{n}\right\}$ represents the set of risk values for all regions.

In summary, this three-stage ensemble learning model comprises three networks: A for extracting regions of interest, B for estimating risk values for each region, and C for aggregating the risk values. The regions of interest are extracted from the input image I using Network A and represented as the set R. Each region $r_{i}$ is then processed by Network B to obtain its corresponding risk value $s_{i}$. Finally, Network C combines all the risk values S to generate the final prediction P. This ensemble learning model effectively performs binary classification tasks.

#### 2.4.1. Network A: ROI Extraction Based on Improved YOLOv5

Network A, a crucial component of the FGDS model, is a target detection network built upon the improved YOLOv5 architecture, which aims to provide fast and accurate real-time object detection [21]. To enhance its detection performance, we have incorporated two advanced techniques: BiFPN (Bi-directional Feature Pyramid Network) and ECA (Efficient Channel Attention).

##### Feature Fusion Network Improvement

In YOLOv5, the feature fusion component relies on PANet, but it falls short in detecting targets with fine-grained details and multi-scale characteristics. To address these limitations, we introduce the BiFPN network, which offers numerous advantages such as multi-scale feature fusion, adaptive feature refinement, enhanced spatial context understanding, scale-aware object representation, and robustness to variations [22]. By integrating BiFPN into the YOLO network, we achieve improved feature fusion, enhanced spatial context understanding, robust handling of scale variations, adaptive feature refinement, and efficient information flow. BiFPN plays a pivotal role in enabling effective feature fusion across multiple scales by combining bottom-up and top-down pathways. This integration seamlessly integrates high-level and low-level features, facilitating precise localization and identification of regions of interest. As a result, our model leverages this hierarchical fusion process to significantly enhance its capability of capturing intricate facial details.

The weight fusion method employed by BiFPN is fast normalization fusion. The formula is as follows:(4)$$O={\sum \;}_{i}\;\frac{w_{i}}{\epsilon +{\sum \;}_{j}\;w_{j}}\cdot I_{i}$$

ϵ = 0.0001; $w_{i}$ greater than or equal to 0 are learnable weight values, with each weight value ranging from 0 to 1; $I_{i}$ representing the feature map.

Using the sixth layer of BiFPN structure (Figure 3) to illustrate its fusion approach:(5)$$P_{6}=\mathrm{Conv}\left(\frac{w_{1}\cdot F_{6}+w_{2}\cdot \mathrm{Resize}\left(F_{7}\right)}{w_{1}+w_{2}+\epsilon }\right)$$
(6)$$N_{6}=\mathrm{Conv}\left(\frac{w_{1}^{’}\cdot F_{6}+w_{2}^{’}\cdot P_{6}+w_{3}^{’}\cdot \mathrm{Resize}\left(N_{5}\right)}{w_{1}^{’}+w_{2}^{’}+w_{3}^{’}+\epsilon }\right)$$

##### Add Attention Mechanism

The attention mechanism is a technique in deep learning that enables models to selectively focus on specific parts of the input data. It assigns different weights or importance to different elements in the input, allowing the model to pay more attention to relevant features and ignore irrelevant or less important ones. The addition of an attention mechanism to YOLOv5 improves localization accuracy, enhances discrimination between objects, enables robust handling of complex scenes, introduces scale-awareness, and enhances generalization capabilities. These advantages collectively result in more accurate and reliable object detection performance.

ECA is a very lightweight and convenient attention module that can significantly improve the performance of networks including ResNets and YOLO, and exhibits better performance results than its counterpart attention modules [23]. Based on this, this study adds the ECA module into the YOLO network.

The simplified mathematical expression for the ECA (Efficient Channel Attention) mechanism is as follows:(7)M=AvgPool⁡(X)
(8)W=FC⁡(ReLU⁡(FC(M)))
(9)Y=X⊙W
(10)Z=Y⋅γ

Here, AvgPool⁡(⋅) represents the global average pooling operation, FC⁡(⋅) represents the fully connected layer operation, ReLU⁡(⋅) represents the *ReLU* activation function; ⊙ represents the element-wise multiplication operation, and γ is a learnable scaling parameter.

### 2.5. Network B: Disease Information Extraction Based on Improved Residual Neural Network

In Network B, each region of interest extracted by Network A corresponds to a convolutional neural network for extracting image features associated with genetic disorders. Each network is trained separately. Considering the complexity of disease information in the region of interest, two convolutional neural network models are designed in this study (CNN A and CNN B), both using the attention mechanism and the residual module to improve model performance, enhance accuracy and avoid overfitting. Due to the non-uniform image size of the ROI region obtained based on object detection, this study used the letterbox method to unify the image size before inputting it into the Network B, unifying the size of all images to 256 × 256.

The CNN A is obtained by adding the CBAM (Convolutional Block Attention Module) attention mechanism to the 34-layer residual network. It is used for disease information extraction on complex regions of interest, such as the chin, nose, jaw, etc. CNN B is obtained by adding an attention mechanism to the 8-layer residual neural network, which is used to extract some simple disease information, such as NT thickness and morphology of the cranial vault. The detailed structures of CNN A and CNN B are shown in Appendix A.

#### 2.5.1. Residuals Module

The residual module was proposed by Kaiming He in the paper titled “Deep Residual Learning for Image Recognition” [24]. It addresses the vanishing gradient problem, facilitates optimization, enables the construction of deeper networks, and promotes gradient flow and feature reuse.

Given an input feature map X, the mathematical expression for a residual module can be represented as:(11)Y=X+F(X)

Here, F(⋅) represents the residual function, which can be a combination of one or multiple convolution operations and nonlinear activation functions.

#### 2.5.2. CBAM Attention Mechanism

By selectively attending to informative channels and spatial locations, the CBAM attention mechanism, consisting of the Channel Attention Module (CAM) and the Spatial Attention Module (SAM), enhances the representational power of CNNs. [25]. This improves feature representation, adaptability to complex patterns, and overall performance in various computer vision tasks.

Given an input feature map X, CBAM consists of two attention mechanisms: CAM and SAM.

CAM:(12)$$A_{c}=\sigma (FC(AvgPool(X)))$$
(13)$$X_{c}=A_{c}⊙X$$

SAM:(14)$$A_{s}=\sigma \left(FC\left(MaxPool\left(X_{c}\right)\right)\right)$$
(15)$$Z=A_{s}⊙X_{c}$$

Here, σ(⋅) represents the sigmoid activation function, FC(⋅) denotes fully connected layers, ⊙ represents element-wise multiplication, and $X_{c}$ represents the feature map after applying channel attention. The resulting feature map Z contains enhanced representations that are more discriminative for downstream tasks.

### 2.6. Network C: Disease Risk Estimation Based on XGBoost Algorithm

Each network undergoes separate training, and the output from the last neural network layer of each network is selected to represent the probability of the genetic disorder. These outputs are then concatenated to form multidimensional features. In our ensemble learning approach, we utilize XGBoost as the stacking model for the sub-networks. XGBoost combines the disease probabilities from sub-networks to estimate the overall probability of a patient having a genetic disorder or not [26].

The training of Networks A, B, and C involved three stages. Initially, Network A was utilized to extract the ROI. Subsequently, CNN A and CNN B were employed to predict the disease probability for each ROI. Finally, the predictions from each network were combined using XGBoost to achieve an overall classification performance.

### 2.7. Data Augmentation

Medical data collected in clinical scenarios exhibits characteristics such as small samples and long-tailed distributions, which are not suitable for training deep learning models. In this study, the data augmentation strategy of Trivial Augment is used to augment the extracted regions of interest and change the sample distribution. Trivial Augment is a simple yet effective data augmentation technique that enhances machine learning models’ performance by generating variations of the original training data [27]. By applying trivial transformations to the text, the method introduces subtle changes that improve the model’s ability to generalize and handle different inputs. The image with data augmentation is shown in Appendix A.

### 2.8. Performance Evaluation

In order to thoroughly assess the remarkable performance of our FGDS model, we conducted training experiments with various alternative models, including well-established CNN architectures such as ResNet [24], VGG [28], and DenseNet [29]. Additionally, to provide a comprehensive analysis of our approach, we compared its performance against other state-of-the-art deep learning models, namely InceptionV3 [30], EfficientNet [22], and Xception [31].

In the statistical analysis, we employed several evaluating indicators to evaluate the screening performance. These indicators include accuracy, sensitivity, specificity, mPA (mean average precision), recall, precision, and F1 score. Moreover, we utilized the receiver operating characteristic (ROC) curve and the area under the ROC curve (AUC) to compare the screening performance across different models and networks.

### 2.9. Heat Map Generation

Grad-CAM is a visualization technique that improves the interpretability of CNN models by incorporating a visualization layer [32]. It utilizes gradients associated with the target concept to produce a localization map, which emphasizes crucial regions in the image responsible for concept prediction. Regions with higher intensity colors indicate more significant predictions.

## 3. Results

### 3.1. Performance of Region of Interest Extraction (Network A)

In Network A, the BiFPN feature fusion network and ECA attention mechanism are used to improve the YOLOv5 network, named YOLOv5 + BiFPN + ECA. After discussion with physicians and reference to clinical data, seven regions of interest in the ultrasound images were extracted, and they were named Chin, head1, head2, D, NA + NB, max + mand and NT.

As shown in Figure 4 and Figure 5 and Table 2, Yolov5 with the enhancements of BiFPN and ECA achieved improved performance compared to the baseline Yolov5. The results for mAP, recall, and precision are as follows: In the “Chin” class, the mAP remained at 0.996, while the recall showed improvement to 0.800 and the precision slightly decreased to 0.702. Similarly, for the “head1” class, the mAP remained at 0.996, with an improved recall of 0.830 and a consistent precision of 0.800. In the case of the “head2” class, the mAP remained at 0.996, with an increased recall of 0.890 and a slightly lower precision of 0.781. For the “D” class, the mAP remained at 0.996, the recall improved to 0.860, and the precision decreased to 0.610. The “NA + NB” class maintained a mAP of 0.992, with unchanged recall and precision values of 0.990 and 0.980, respectively. Likewise, the “max + mand” class maintained a mAP of 0.996, with a consistent recall of 0.990 and a slightly decreased precision of 0.712. The “NT” class showed an improved mAP of 0.950, while recall and precision remained at 0.980. Lastly, for all classes combined, the mAP remained at 0.986, with a steady precision of 0.988.

### 3.2. Performance of Disease Information Extraction (Network B)

In Network B, our research establishes associations between diseases and anatomical structures by matching regions of interest with disease labels, obtaining risk values for each region of interest. In this paper, the performance of Network B was tested in a retrospective dataset, and the contribution of each region of interest to screening for genetic disorders was as follows:

The Table 3 and Figure 6 represents the screening performance of different ROI areas for genetic disorder detection. The metrics evaluated include AUC, sensitivity, specificity, accuracy, and F1 score. For the “chin” ROI, the AUC is 0.833, indicating a reasonably good performance. The sensitivity is 0.594, suggesting that it correctly identifies positive cases around 59.4% of the time. The specificity is high at 0.930, implying a low false-positive rate. The accuracy is 0.786, representing the overall correctness of the predictions. The F1 score, which combines precision and recall, is 0.704. Similarly, for other ROIs, such as “head1,” “head2,” “max + mand,” “NA + NB,” and “NT,” the corresponding metrics are provided. These include AUC, sensitivity, specificity, accuracy, and F1 score. From the table, we can observe that different ROIs exhibit varying levels of screening performance. For instance, “head2” has a relatively higher AUC of 0.764 and sensitivity of 0.576, indicating better discrimination and a higher ability to correctly identify positive cases compared to the other ROIs. On the other hand, “max + mand” has a lower AUC of 0.589 and sensitivity of 0.438, suggesting a comparatively weaker performance in detecting genetic disorders.

Based on these results, it can be inferred that certain ROIs might have a stronger association with specific genetic disorders, leading to better screening performance. Further analysis and investigation of these ROIs and their relationship with genetic disorders could provide valuable insights for disease detection and diagnosis.

### 3.3. Performance of Genetic Disorder Prediction in the FGDS Model (Internal Test Set)

The Table 4 and Figure 7a presents the performance metrics of different models, where FGDS represents a genetic disorder screening model developed in the research, while the remaining models serve as baselines.

In terms of performance, FGDS achieves an AUC of 0.903, indicating excellent discrimination ability. It demonstrates a sensitivity of 0.753, which suggests that it correctly identifies 75.3% of positive cases. The specificity of FGDS is 0.889, indicating a relatively low false-positive rate. Furthermore, the F1 score of 0.790 reflects a balanced precision and recall. DenseNet-201 achieves an AUC of 0.845 with a sensitivity of 0.763. It maintains a relatively high specificity of 0.807, resulting in a balanced performance with an F1 score of 0.756; ResNet-34 shows an AUC of 0.827, a sensitivity of 0.655, and a high specificity of 0.888. However, the F1 score is slightly lower at 0.727 compared to the previous models; VGG-16 achieves an AUC of 0.786 with a sensitivity of 0.633. The specificity is 0.833, indicating a relatively low false-positive rate. The F1 score is 0.683, reflecting a trade-off between precision and recall; InceptionV3 demonstrates an AUC of 0.853 and a sensitivity of 0.712. It maintains a specificity of 0.867, resulting in an F1 score of 0.754; EfficientNetB1 exhibits the lowest AUC of 0.738 and a sensitivity of 0.497. However, it demonstrates a high specificity of 0.914, resulting in an F1 score of 0.617; Xception achieves an AUC of 0.834 with a sensitivity of 0.638. It shows a high specificity of 0.953, resulting in an F1 score of 0.751.

From the table, it can be observed that FGDS has the highest AUC and relatively balanced sensitivity and specificity, indicating its strong overall performance in disease detection. Other models, such as DenseNet-201 and InceptionV3, also exhibit a favorable performance.

### 3.4. Performance of Genetic Disorder Prediction in the FGDS Model (External Test Set)

In deep learning model performance testing, the use of an external test dataset plays a crucial role in evaluating the model’s generalization and robustness. In this experiment, we prospectively collected an external test set to evaluate the model’s performance.

As shown in Figure 7b and Table 5, In the prospective test set, FGDS achieves an AUC of 0.844, indicating a good discriminative ability in distinguishing between positive and negative cases. It exhibits a sensitivity of 0.768, meaning it correctly identifies 76.8% of positive cases. The specificity of FGDS is 0.837, suggesting a relatively low false-positive rate. Furthermore, the F1 score of 0.806 reflects a balanced precision and recall. In comparison, DenseNet-201 achieves an AUC of 0.717, a sensitivity of 0.556, a specificity of 0.759, and an F1 score of 0.632. InceptionV3 shows an AUC of 0.733, a sensitivity of 0.778, a specificity of 0.590, and an F1 score of 0.733. Based on these results, it is evident that the FGDS model outperforms DenseNet-201 and InceptionV3 in terms of AUC, specificity, and F1 score. FGDS exhibits higher discriminative ability, correctly identifies a larger proportion of positive cases, and maintains a more balanced precision and recall trade-off. Moreover, FGDS achieves a higher specificity compared to InceptionV3, indicating a lower false-positive rate.

On the other hand, ResNet-34 exhibits relatively low performance with an AUC of 0.568, a sensitivity of 0.308, a specificity of 0.801, and an F1 score of 0.418. VGG-16 demonstrates low AUC (0.429), specificity (0.151), and F1 score (0.701), while EfficientNetB1 and Xception also demonstrate limitations in various performance metrics.

### 3.5. The Heat Map of the FGDS Model

From Figure 8, it is evident that the forehead, mouth, and nose regions make significant contributions to the screening of genetic disorders. Additionally, FGDS appears to be highly sensitive to fetal facial contours, as it focuses on contour information in ROI regions such as “NA + NB,” “chin,” and “head1.” In the “NT” region of interest, the FGDS model unsurprisingly pays attention to the thickness information of the nuchal translucency, which is an important clinical indicator for genetic disorder screening.

## 4. Discussion

In this study, we have developed a novel three-stage ensemble model called FGDS for screening genetic disorders in fetuses. The FGDS model has demonstrated its effectiveness in detecting genetic disorders, surpassing ResNet, DenseNet, InceptionV3, and other algorithms. Through Network A, we have discovered that the deep learning-based target detection technique enables the accurate extraction of anatomical structures from ultrasound images of fetal median sagittal sections. In our comparative analysis of Network B, we have identified that the model developed using the “Chin” region has achieved the highest performance. Furthermore, based on our interpretable experiments, we have found that the mouth, NT, and nose regions play a significant role in the manifestation of genetic disorders on the face.

Recent genetic studies have established a significant correlation between facial abnormalities in individuals with genetic disorders and specific gene mutations [33,34]. For instance, genes such as 10q25.3, 8q24, VAX1, IRF6, and others are associated with cleft lip disease and impact the development of the human jaw and maxilla, leading to abnormalities in various facial features such as the nasal wing, cheek, and lips. Certain genetic loci, including Rs287104 in the KCTD15 gene, Rs9995821 in the DCHS2 gene, Rs2977562 in the 3q21.3 gene, and Rs10176525 in the 2q36.1 gene, are also linked to specific facial traits such as the nasal tip and alar morphology, nostril aperture, upper lip thickness, and nasal bridge height [35,36,37,38].

Our study utilizes object detection technology in Network A of FGDS to extract regions of interest in ultrasound images, eliminating interference caused by irrelevant information such as noise and artifacts. The decision layer fusion method based on meta learning for regions of interest not only emphasizes the contribution of a single region of interest to the disease, but also establishes the interaction relationships between different regions of interest, effectively improving the robustness of the FGDS model. Due to the unique imaging method of ultrasound imaging, its imaging quality is greatly influenced by factors such as instrument type, external environment, and the doctor’s experience. If the model structure is not customized for specific tasks, it will be difficult to obtain robust prediction models. This is also why mainstream deep learning models can achieve significant classification performance in internal test sets, but are poor in external test sets. Additionally, the data distribution in this study follows a long-tail distribution commonly observed in medical data, where positive samples are significantly fewer than negative samples. To address this issue, the study employed data augmentation techniques to modify the sample distribution in the training set, thereby increasing the complexity of positive samples and effectively improving the model’s performance.

Deep learning has often been considered a high-risk approach in medical decision-making due to its lack of interpretability. To address this concern, we utilized the Grad-CAM to generate heatmaps, highlighting the anatomic structure that significantly contribute to the model’s classification. These heatmaps effectively pinpointed areas which played a crucial role in the classification process. Importantly, we also presented the heatmaps of individual subgraphs within the model, providing a comprehensive understanding of the evaluation basis from different perspectives. Network A also demonstrates that FGDS accurately recognizes the anatomical structure of the fetal face, achieving an average precision of 0.988 across all classes. As illustrated in Figure 7 and Figure 8, the “head2” region of interest in Network B displayed heightened sensitivity to the middle and lower face, while the “chin” exhibited a finer depiction of the development of the nasal bone, jaw bone, and other facial components. “head1”, on the other hand, showed greater sensitivity to the frontal bone and hindbrain. This interpretable feature holds potential for its integration into complementary medicine, facilitating a more comprehensive understanding and evaluation of complex cases.

In the field of clinical medicine, the potential applications of the proposed technology in the future are multifaceted and hold great promise. Firstly, the development of low-cost screening software that can be readily packaged and deployed on cloud or server platforms would offer a convenient and cost-effective approach to facilitate early detection of genetic disorders. This approach has the potential to revolutionize healthcare delivery by enabling broader accessibility and increasing the reach of screening programs, particularly in resource-limited settings. By leveraging the power of cloud computing and server infrastructure, healthcare professionals would be able to utilize this software to efficiently analyze large volumes of patient data, facilitating prompt identification of individuals at risk and enabling timely interventions and treatments. Secondly, the deployment of this technology in hospitals at all levels can greatly contribute to graded diagnosis and treatment strategies, which are aimed at reducing medical errors and minimizing medical waste. With its ability to provide accurate and reliable diagnostic insights, the technology can aid healthcare providers in making informed decisions. This personalized approach can help optimize patient outcomes and improve the overall efficiency of healthcare systems, ensuring that resources are utilized effectively. With its ability to analyze complex ultrasound image data and generate interpretable results, it can provide healthcare professionals with a comprehensive understanding of a patient’s genetic condition, potentially uncovering critical insights that might have been missed through conventional diagnostic methods. By incorporating this technology into the diagnostic workflow, doctors can benefit from its assistance in formulating accurate and timely diagnoses, thereby enhancing the overall diagnostic accuracy and clinical decision-making process. Finally, the scalability of the technology lends itself to large-scale, cross-regional genetic disorder screening initiatives. By leveraging its computational power and capacity to analyze vast amounts of genetic data, the technology can facilitate systematic screening programs across diverse populations and geographical regions. This can enable the identification of population-specific genetic variations and the early detection of genetic disorders on a broader scale. Such large-scale screening initiatives can inform public health policies, facilitate the implementation of targeted prevention strategies, and contribute to the advancement of precision medicine by enabling the identification of rare genetic disorders that may have previously gone undetected.

While the current study highlights the potential of our FGDS model, it is important to acknowledge its limitations. Firstly, the study sample was limited, which may impact the generalizability of the model, particularly across different racial and ethnic populations such as African descent, Latinos, Caucasians, etc. To enhance the model’s robustness, our next study will involve data collection from diverse racial backgrounds, ensuring broader representation and applicability. Secondly, although this study included various genetic disorder cases, many rare genetic disorders with distinct facial abnormalities were not included. Future investigations will entail collaborating with multiple institutions to collect these rare cases, thus expanding the dataset and capturing a more comprehensive range of genetic disorders and associated facial features. Furthermore, this study solely employed a single standard section of ultrasonic images for genetic disorder screening. Subsequent studies will explore the utilization of other standard sections, thereby enabling a more comprehensive and multi-dimensional analysis of facial characteristics in the context of genetic disorders. Additionally, while the current model effectively serves as a screening tool, it cannot identify genetic disease types. This study is not immediately available for clinical use and social factors (ethics and morality) need to be considered for clinical application. Our future focus lies in gathering a larger collection of ultrasonic facial images for each genetic disorder category and subsequently developing an AI model capable of providing specific diagnoses. Moreover, this experiment did not include a comparison between our model and integrated tests or NIPT. To address this gap, we plan to conduct this experiment in the next research, allowing for a comprehensive evaluation of the model’s performance and its potential integration with existing diagnostic approaches. In terms of technology, the ROI extraction performance of Network A needs improvement for certain categories, particularly “Max + Mand” and “head2”. In future studies, we will consider using more advanced object detection methods to enhance the performance of Network A. In Network B, we can utilize some state-of-the-art classification models, such as Transformer. In future research, we will attempt to experiment with such models and continuously improve the classification performance of our model.

## 5. Conclusions

In summary, the current study successfully constructed a novel three-stage ensemble learning model called FGDS, which facilitates automated screening of genetic disorders during the prenatal stage and generates informative heat maps. Experimental results demonstrate that FGDS accurately recognizes fetal facial anatomical structures, such as the nose, jaw, and forehead. These specific facial features provide crucial diagnostic information for identifying fetal genetic disorders. By assisting in prenatal ultrasound diagnosis, this framework has the potential to reduce false-negative results and address the scarcity of medical resources, thereby enhancing the overall effectiveness of genetic disorder screening in prenatal settings.

## Figures and Tables

**Figure 1 bioengineering-10-00873-f001:**
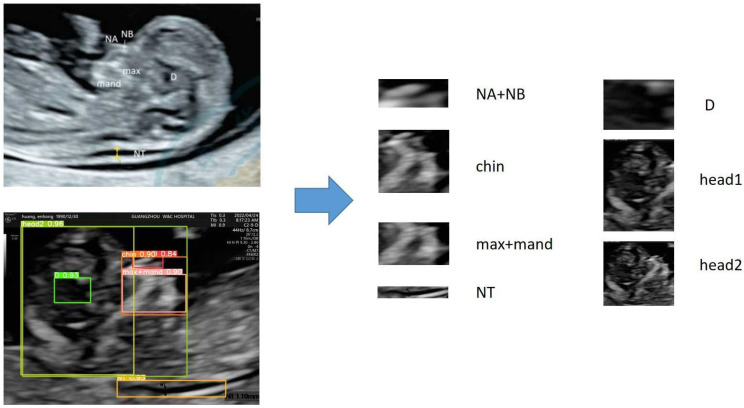
Determination of regions of interest (ROI). NA + NB: apex nasi and nasal bone; NT: nuchal translucency; D: Diencephalon.

**Figure 2 bioengineering-10-00873-f002:**
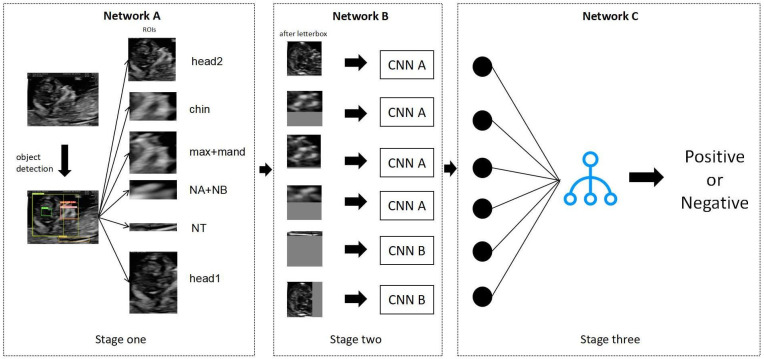
Model Architecture. The utilization of the FGDS model encompasses three distinct stages, each serving a specific purpose. In the initial stage, the image undergoes preprocessing through homomorphic filtering to enhance its quality. The preprocessed image is then fed into Network A, which focuses on extracting the relevant regions of interest from the image. Moving on to the second stage, the extracted regions of interest are resized to a standardized dimension of 256 × 256. These resized regions are subsequently inputted into Network B, which is responsible for extracting disease-related information from each individual region of interest. Finally, in the third stage, the low-dimensional vectors obtained from each region of interest are concatenated together. This concatenated vector is then fed into the XGBoost algorithm, which performs the final classification task, yielding the predicted outcome for the genetic disorder under consideration. The following provides a comprehensive and detailed overview of the technical methodologies employed within each section of the FGDS model.

**Figure 3 bioengineering-10-00873-f003:**
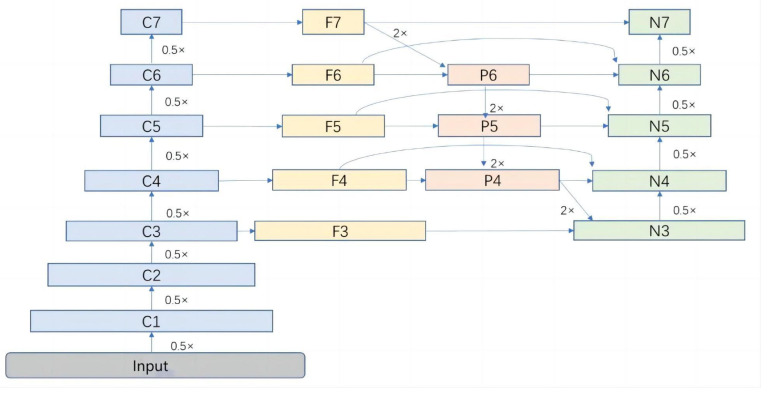
BiFPN network architecture.

**Figure 4 bioengineering-10-00873-f004:**
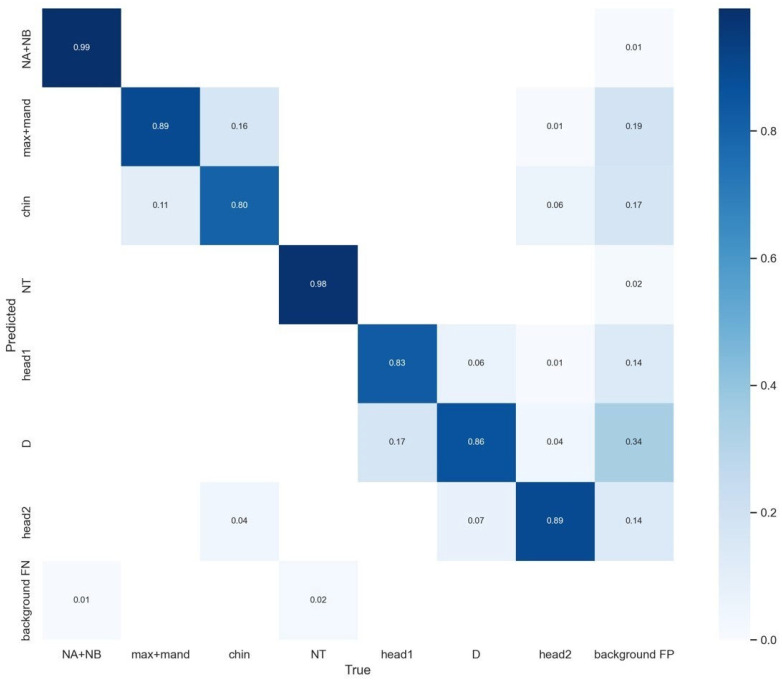
Confusion matrix of Network A.

**Figure 5 bioengineering-10-00873-f005:**
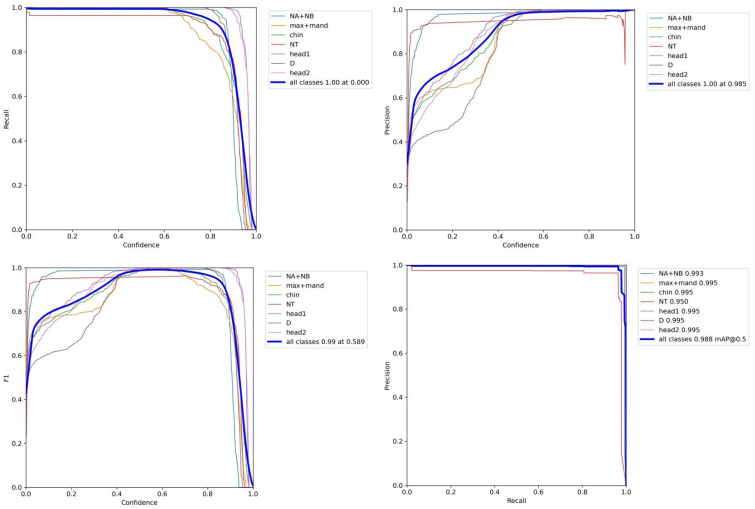
Recall-Confidence, Precision-Confidence, F1-Confidence, and P-R curves.

**Figure 6 bioengineering-10-00873-f006:**
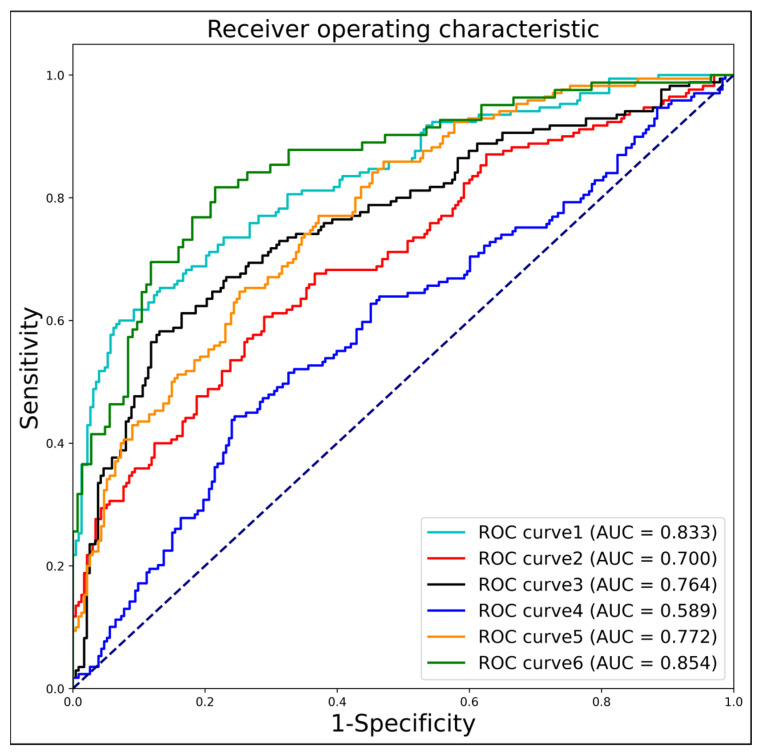
Network B: Performance of disease information extraction. As shown in Figure 3 and Table 3, curves 1 to 6 represent six regions of interest, namely “chin”, “head1”, “head2”, “max + mand”, “NA + NB”, and “NT”. The dotted line represents the ROC curve of a completely random classifier.

**Figure 7 bioengineering-10-00873-f007:**
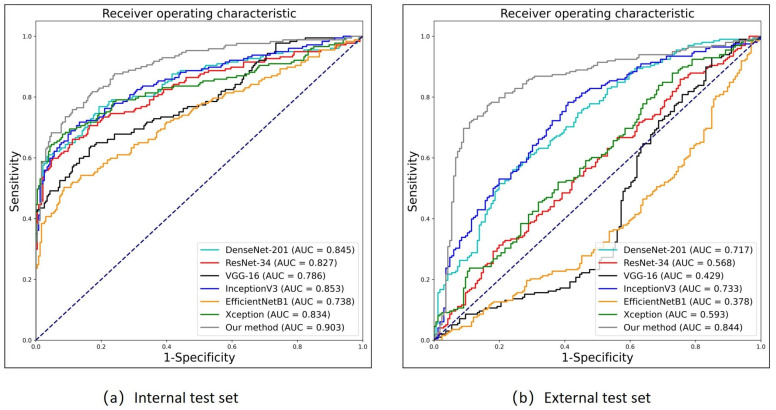
Performance of genetic disorder prediction in the FGDS model. (**a**) Internal test set. (**b**) External test set. “Our method” represents the FGDS model developed in this study. The dotted line represents the ROC curve of a completely random classifier.

**Figure 8 bioengineering-10-00873-f008:**
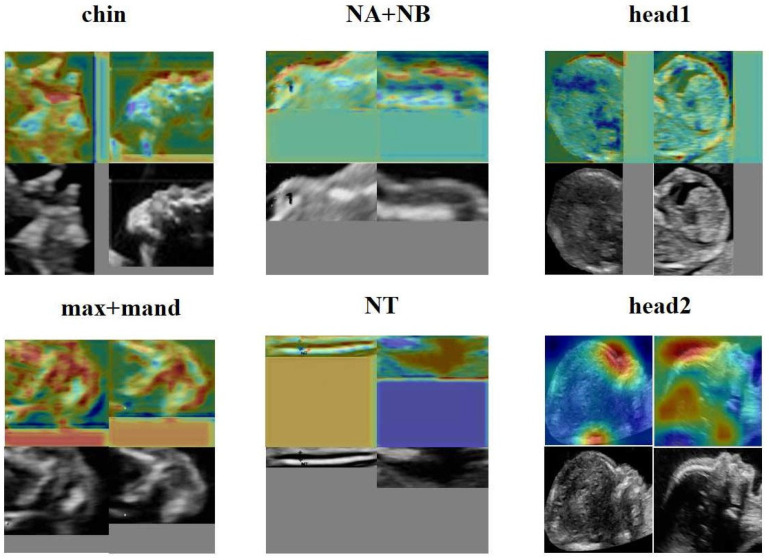
Interpretable heat map obtained from Network B.

**Table 1 bioengineering-10-00873-t001:** Dataset characteristics.

	Retrospective Dataset		Prospective Dataset
	Training Dataset	Internal Test Set	External Test Set
All images	1780	410	364
Images of negative results	1094	233	166
Images of positive results	686	177	198
Ultrasound equipment	GE Volution E10		GE Voluson E6/E8/E10, Philips iE33
Ultrasonographer	Four ultrasonographers		More than 10 ultrasonographers

**Table 2 bioengineering-10-00873-t002:** Object detection performance of Network A.

	Yolov5			YOLOv5 + BiFPN + ECA(Network A)		
	mAP	Recall	Precision	mAP	Recall	Precision
Chin	0.996	0.740	0.771	0.995	0.800	0.702
head1	0.996	0.660	0.680	0.995	0.830	0.800
head2	0.996	0.980	0.883	0.995	0.890	0.781
D	0.996	0.850	0.552	0.995	0.860	0.610
NA + NB	0.992	0.990	0.980	0.993	0.990	0.990
max + mand	0.996	0.990	0.723	0.995	0.890	0.712
NT	0.932	0.950	0.969	0.9500	0.980	0.980
all classes	0.986			0.988		

**Table 3 bioengineering-10-00873-t003:** Network B: Performance of disease information extraction.

ROI	Curve Name	AUC	Sensitivity	Specificity	Accuracy	F1
chin	curve1	0.833	0.594 (0.516–0.668)	0.930 (0.886–0.958)	0.786	0.704
head1	curve2	0.700	0.600 (0.522–0.673)	0.711 (0.647–0.767)	0.664	0.600
head2	curve3	0.764	0.576 (0.498–0.651)	0.869 (0.818–0.908)	0.747	0.656
max + mand	curve4	0.589	0.438 (0.362–0.516)	0.755 (0.694–0.808)	0.622	0.493
NA + NB	curve5	0.772	0.765 (0.692–0.825)	0.628 (0.562–0.690)	0.686	0.672
NT	curve6	0.854	0.805 (0.700–0.881)	0.785 (0.707–0.847)	0.792	0.737

**Table 4 bioengineering-10-00873-t004:** Screening performance of the FGDS in internal test sets.

Model	AUC	Sensitivity	Specificity	F1
FGDS	0.903	0.753 (0.680–0.814)	0.889 (0.839–0.924)	0.790
DenseNet-201	0.845	0.763 (0.692–0.822)	0.807 (0.749–0.854)	0.756
ResNet-34	0.827	0.655 (0.580–0.724)	0.888 (0.839–0.924)	0.727
VGG-16	0.786	0.633 (0.557–0.703)	0.833 (0.777–0.877)	0.683
InceptionV3	0.853	0.712 (0.638–0.776)	0.867 (0.815–0.906)	0.754
EfficientNetB1	0.738	0.497 (0.422–0.573)	0.914 (0.869–0.945)	0.617
Xception	0.834	0.638 (0.563–0.708)	0.953 (0.915–0.975)	0.751

**Table 5 bioengineering-10-00873-t005:** Screening performance of the FGDS in external test sets.

Model	AUC	Sensitivity	Specificity	F1
FGDS	0.844	0.768 (0.701–0.823)	0.837 (0.770–0.888)	0.806
DenseNet-201	0.717	0.556 (0.483–0.625)	0.759 (0.685–0.820)	0.632
ResNet-34	0.568	0.308 (0.246–0.378)	0.801 (0.731–0.857)	0.418
VGG-16	0.429	0.924 (0.876–0.955)	0.151 (0.102–0.216)	0.701
InceptionV3	0.733	0.778 (0.712–0.832)	0.590 (0.511–0.665)	0.733
EfficientNetB1	0.378	0.975 (0.939–0.991)	0.030 (0.011–0.072)	0.699
Xception	0.593	0.833 (0.772–0.881)	0.313 (0.245–0.390)	0.692

## Data Availability

The data generated and/or analyzed during the current study are available upon reasonable request from the corresponding author. The data can be accessed only for research purposes. Researchers interested in using our data must provide a summary of the research they intend to conduct. The reviews will be completed within 2 weeks and then a decision will be sent to the applicant. The data are not publicly available due to hospital regulation restrictions.

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
