# Peer review of "An Innovative Three-Stage Model for Prenatal Genetic Disorder Detection Based on Region-of-Interest in Fetal Ultrasound"

_bioengineering, 2023, doi:10.3390/bioengineering10070873_

Round 1
Reviewer 1 Report
The authors propose a method for detecting fetal facial deformities caused by genetic factors. Their proposed method is interesting from a machine learning perspective, the dataset is well prepared, and sufficient numerical experiments have been conducted. The paper is generally well written.
From a scientific perspective, this study is well done, and from an ethical perspective, it has passed the ethics committee. So, as a reviewer, I believe this study is worthy of publication.
However, let me express one concern.
Once I reviewed a research study on a method to detect facial deformities caused by genetic factors using ultrasound imaging, but the study was rejected due to very strong protests from a reviewer who seems to have been a clinician. In general, this research topic is very sensitive in the western research community. Of course, I understand that there are communities with different social environments. However, I hope that the authors will include in the limitation and discussion that this study is not immediately available for clinical use and that the social factors need to be considered for clinical application.
Author Response
Dear reviewer 1,
Thank you for your invaluable support of our research. We sincerely appreciate your insightful perspective. It is important to acknowledge that in Western contexts, this method may encounter certain limitations due to variations in social environments. Taking into consideration your valuable suggestion, we have made sure to address and incorporate this aspect within our discussion on limitations.
“ This study is not immediately available for clinical use and that the social factors (ethics and morality) need to be considered for clinical application.” (Line 558-560)
Reviewer 2 Report
Authors has shown a nice research work in this manscript. Please find my comments below.
1. In section 2. the gold standard needs a reference. Four main categories need detail.
2. Mean and variance of subjects should be provided.
3. Abbreviation in Figure 1 should be provided. Are the label boxes are generated by the ultrasound equipments?
4, Class imbalance presents in dataset. A discussion will be provied for justification of the findings.
5. MAX+Mnad class is the worst in performance. A separated discussion and possible future work for impovment should be provided.
6. Transformer is the state-of-art in deep learning. Future work should include the possible use of this technquie and other foundation model.
Quality is fine.
Author Response
Dear reviewer 2,
Thank you for your support of our research. We have made substantial improvements based on your suggestions. Below is our response to your feedback.
- In section 2. the gold standard needs a reference. Four main categories need detail.
R: We have added references and provided detailed descriptions for four categories.
All cases have genetic test results, which are used as the gold standard for diagnosis [17]. (Line 99)
[17] Greely H T. Get ready for the flood of fetal gene screening. Nature. 2011, 469(7330): 289-291.
Four main categories is T21、T18、T13、Rare genetic disorder.
Rare genetic disorders: Include Turner syndrome, 1q21.1 microdeletion, and 15q11-q13 duplication, Helsmoortel-van der AA syndrome, 15q26.1-q26.3 deletion, and 20p13 duplication and other Monogenic Genetic disorders.
T21: Down syndrome (Line 51-52)
T18: Trisomy 18 syndrome (Line 51-52)
T13: Trisomy 13 syndrome (Line 51-52)
- Mean and variance of subjects should be provided.
R: The majority of participants in this study were retrospective data, as medical information technology was not widely available at that time, and we were unable to retain gestational age of the participants in the retrospective data. In the prospective data, we calculated the gestational age of the participants as 14.8 ± 2.6.
- Abbreviation in Figure 1 should be provided. Are the label boxes are generated by the ultrasound equipments?
R: We added explanations for abbreviations. Figure 1. Determination of regions of interest (ROI). NA+NB: apex nasi and nasal bone; NT: nuchal translucency; D: Diencephalon. (Line 147-148).
In this study, a region of interest extraction network was designed in network A. The label boxes were generated by this region of interest extraction network. The labels for training this network were annotated by medical professionals. (Figure 2. Model Architecture )
- Class imbalance presents in dataset. A discussion will be provied for justification of the findings.
R: In the discussion, we included a discussion about the data distribution.
“Additionally, the data distribution in this study follows a long-tail distribution commonly observed in medical data, where positive samples are significantly fewer than negative samples. To address this issue, the study employed data augmentation techniques to modify the sample distribution in the training set, thereby increasing the complexity of positive samples and effectively improving the model's performance.”(Line 490-495)
- MAX+Mand class is the worst in performance. A separated discussion and possible future work for impovment should be provided.
R: The ROI extraction performance of network A needs improvement for certain categories, particularly “Max+Mand” and “head2”. In future studies, we will consider using more advanced object detection methods to enhance the performance of network A. (Line 566-569)
- Transformer is the state-of-art in deep learning. Future work should include the possible use of this technquie and other foundation model.
R: In Network B, we can utilize some state-of-the-art classification models, such as Transformer. In future research, we will attempt to experiment with such models and continuously improve the classification performance of our model. (Line 569-572)